# A Tight-Connection *g*-C_3_N_4_/BiOBr (001) S-Scheme Heterojunction Photocatalyst for Boosting Photocatalytic Degradation of Organic Pollutants

**DOI:** 10.3390/nano14131071

**Published:** 2024-06-22

**Authors:** Xinyi Zhang, Weixia Li, Liangqing Hu, Mingming Gao, Jing Feng

**Affiliations:** 1Key Laboratory of Superlight Materials & Surface Technology of Ministry of Education, Harbin Engineering University, Harbin 150001, China; zxy0509@hrbeu.edu.cn (X.Z.); fh013876@china-fuhai.com (W.L.); hlqing@hrbeu.edu.cn (L.H.); 2Qilu Institute of Technology, College of Biological and Chemical Engineering, Jinan 250200, China

**Keywords:** *g*-C_3_N_4_/BiOBr, S-scheme heterojunction, photocatalysis, exposed crystal plane

## Abstract

The efficient separation of photogenerated charge carriers and strong oxidizing properties can improve photocatalytic performance. Here, we combine the construction of a tightly connected S-scheme heterojunction with the exposure of an active crystal plane to prepare *g*-C_3_N_4_/BiOBr for the degradation of high-concentration organic pollutants. This strategy effectively improves the separation efficiency of photogenerated carriers and the number of active sites. Notably, the synthesized *g*-C_3_N_4_/BiOBr displays excellent photocatalytic degradation activity towards various organic pollutants, including methylene blue (MB, 90.8%), congo red (CR, 99.2%), and tetracycline (TC, 89%). Furthermore, the photocatalytic degradation performance of *g*-C_3_N_4_/BiOBr for MB maintains 80% efficiency under natural water quality (tap water, lake water, river water), and a wide pH range (pH = 4–10). Its excellent photocatalytic activity is attributed to the tight connection between *g*-C_3_N_4_ and BiOBr in the S-scheme heterojunction interface, as well as the exposure of highly active (001) crystal planes. These improve the efficiency of the separation of photogenerated carriers, and maintain their strong oxidation capability. This work presents a simple approach to improving the separation of electrons and holes by tightly combining two components within a heterojunction.

## 1. Introduction

In recent years, the rapid development of industrialization has unavoidably discharged a large amount of wastewater containing organic pollutants, such as dyes, antibiotics, and pesticides, causing health hazards [1,2,3]. Especially high concentrations of pollutants can lead to more serious health hazards. Notably, high concentrations of pollutants can lead to more serious health pitfalls. According to reports in the literature, the wastewater from the production of antibiotics still contains high concentrations of antibiotic residues and has been identified as an important source of antibiotic contamination [4]. Among them, the main source of high-concentration TC is concentrated wastewater discharged by manufacturers [5]. The wastewater produced from concentrated antibiotic production mainly comes from high-concentration organic wastewater that is separated, extracted, and refined, such as waste mother liquor, waste acid water, etc., which is difficult-to-treat wastewater. Dye production wastewater is diluted before discharge, but the dye concentration in the treated wastewater remains at 100–500 mg L^−1^ [6]. High-concentration organic pollutants are generally difficult to degrade, and traditional physical, chemical, or biological treatment methods often struggle to achieve ideal treatment results. For example, advanced oxidation technologies (such as the Fenton method and ozone oxidation method) can achieve the effective degradation of organic pollutants, but their operating costs are high and difficult to promote and apply. Photocatalytic technology using semiconductors for degrading organic pollutants in wastewater is considered as a promising strategy [7,8,9].

Graphite carbon nitride (*g*-C_3_N_4_) is a typical two-dimensional non-metallic catalyst, presenting suitable energy band positions, a relatively narrow band gap, a unique electronic structure, and stable physical and chemical properties [10]. The positions of the band gaps in *g*-C_3_N_4_ and BiOBr appropriately match (VB*_g_*_-C_3_N_4__ ≈ 1.59 V, CB*_g_*_-C_3_N_4__ ≈ −0.94 V), allowing them to construct a S-scheme heterojunction. Due to having a more negative conduction band (CB) than E_0_ (O_2_/O_2_^−^) (−0.33 V), the photogenerated electrons of *g*-C_3_N_4_ possess the reducing capacity to generate superoxide radicals (·O_2_^−^), which participate in the degradation process of organic pollutants.

Nowadays, BiOBr has attracted much attention for photocatalytic degradation due to its suitable band gap (2.7–2.9 eV) and unique structures [11,12]. The tetragonal structure of BiOBr consists of alternating the stacking of [Bi_2_O_2_]^2+^ layers and double Br^−^ layers, with strong intra-layer covalent bonds and weak inter-layer force [13,14,15,16]. This layer structure provides the space for polarization to improve the separation of photogenerated carriers [17,18,19]. For instance, Zuo et al. synthesized BiOBr/Mn-Ti_3_C_2_T_x_ through in situ ion modification, and the synergistic effect of surface modification and built-in electric field construction effectively improved the carrier separation efficiency and electron utilization efficiency [20]. Zhen et al. optimized the band structure of BiOBr by doping N while introducing abundant oxygen vacancies, resulting in a degradation efficiency of 99.7% for sulfamethoxazole (SIZ) [21]. Yang et al. discovered that S-doping BiOBr enhanced the polarization between the [Bi_2_O_2_S]^2+^ and [Br_2_]^2−^ layers, resulting in the efficient separation of photogenerated carriers [15].

According to the photocatalytic mechanism, the photocatalytic performance of BiOBr can be promoted by improving the carrier separation efficiency and enhancing surface activity. Currently, the effective methods to improve the photocatalytic efficiency include noble metal modification (Pt, Au, Ag, etc.) [22,23,24,25], crystal plane engineering (exposure of active crystal surface) [26,27,28,29], and heterojunction construction [30,31,32,33]. Among them, the S-scheme heterojunction separates photogenerated carriers and maintains a strong redox ability effectively [34,35].

In 2019, Yu et al. first proposed a stepped scheme charge transfer method based on the traditional Z-scheme charge transfer process to describe the enhanced activity of WO_3_/g-C_3_N_4_ systems [36]. Commonly, a S-scheme heterojunction catalyst combined an oxide semiconductor (OP) with a reduction photocatalyst (RP) [37]. After contract, a built-in electric field (IEF) is formed, which can be attributed to the difference in Fermi energy levels (*E_f_*) [38,39]. The IEF from RP to OP effectively promotes the migration of charge carriers [40]. The reserved holes in the positive valence band (VB) allow BiOBr to be an oxidative photocatalyst in the S-scheme heterojunction.

Furthermore, the modulation of the crystal surface generally is considered to optimize the surface atomic arrangement and surface chemistry. Thus, the crystal surface is a critical factor to enhance the photocatalytic activity of a semiconductor. Sun et al. reported that the exposure of the (001) crystal plane in BiOBr increased the active sites, thereby improving photocatalytic efficiency [41].

Here, we combine the exposure of the active crystal plane with S-heterojunctions to enhance the photocatalytic performance of BiOBr. The exposure of the (001) crystal plane provides more active sites. The tightly connected S-scheme heterojunction between BiOBr and *g*-C_3_N_4_ was observed to provide a channel for electron transfer. The S-scheme heterojunction helps to separate the carriers and retains the strong redox ability of e^−^ and h^+^ to degrade organic pollutants. This study provides a strategy for constructing tight structures in S-scheme heterojunction.

## 2. Experimental

### 2.1. Catalyst Preparation

#### 2.1.1. Preparation of Bulk *g*-C_3_N_4_

The bulk *g*-C_3_N_4_ was prepared through pyrolysis of melamine in air. Specifically, melamine (AR, 5 g) was grinded thoroughly in a mortar, and heated at 500 °C for 4 h with a heating rate of 3 °C min^−1^. The bulk *g*-C_3_N_4_ was obtained after grinding.

#### 2.1.2. Preparation of x-*g*-C_3_N_4_/BiOBr

The x-*g*-C_3_N_4_/BiOBr was synthesized by a hydrothermal method. The KBr (AR, 0.3570 g) was added in the solution of Bi(NO_3_)_3_·6H_2_O (AR, 70 mL, 20.79 g L^−1^), and stirred for 2 h to obtain a white suspension. According to the weight ratios of the obtained BiOBr/*g*-C_3_N_4_ (10:4, 10:6, 10:8), the *g*-C_3_N_4_ (0.3657 g, 0.5488 g, 0.7317 g) was dispersed in the above white suspension with stirring for 30 min. Then, this suspension was transferred into a kettle (100 mL) and heated at 160 °C for 12 h. The obtained precipitate was washed three times using deionized water and ethanol. Eventually, the x-*g*-C_3_N_4_/BiOBr was collected by drying at 60 °C for 6 h and full grinding. The collected samples were labeled as 4-*g*-C_3_N_4_/BiOBr, 6-*g*-C_3_N_4_/BiOBr (*g*-C_3_N_4_/BiOBr), and 8-*g*-C_3_N_4_/BiOBr.

In addition, the physical mixed samples were for comparison. The physical mixed sample was obtained by grinding the above *g*-C_3_N_4_ (0.5488 g) and BiOBr (0.9147 g) for 10 min.

The preparation of pure BiOBr follows the same steps as above without the addition of *g*-C_3_N_4_.

### 2.2. Characterization

The crystal structure and phase composition of the prepared samples are characterized by X-ray diffraction (XRD, TTR-III) with Cu-Kα radiation under 40 kV and 150 mV. Fourier transform infrared (FT-IR) spectroscopy is used to characterize the vibrations of functional groups on the catalyst surface. The microscopic morphology of the samples is observed and analyzed by scanning electron microscopy (SEM, SU70-HSD) and transmission electron microscopy (TEM, JEM2010). The thermogravimetric analysis (TGA) curve is measured by a differential thermal analyzer (NETZSCH STA 449 F3), which reflects the relationship between the quality of the reaction material and temperature or time. X-ray photoelectron spectroscopy is utilized to identify the chemical valence states of different elements. The UV-Vis absorption spectra are measured by UV-vis spectrophotometer (UV-2450). Electrochemical experiments are investigated using a three-electrode system (on CHI 660E electrochemical workstation). The photoluminescence (PL) spectra are measured at the excitation wavelength of an Edinburgh FLS 980 instrument.

### 2.3. Photocatalytic Degradation Experiments

The photocatalytic activity of the photocatalysts was evaluated by the visible-light-induced photodegradation reaction of an organic pollutant (MB, CR, TC) in aqueous solution. A 300 W Xe lamp (λ > 420 nm) was used as the experimental light source for the whole photodegradation reaction. Briefly, 0.05 g of catalyst was put into 100 mL of pollutant (MB (20 mg L^−1^), CR (20 mg L^−1^), and TC (50 mg L^−1^)) solution. Prior to the light exposure, the suspension was stirred continuously for 30 min under dark conditions to reach the absorption–desorption equilibrium. During the light exposure, a certain amount of the contaminant solution was taken from the beaker every 10 min, the upper clear solution was taken after centrifugation, and the absorbance of the filtrate was analyzed by UV-vis spectrophotometer. Calculation of degradation efficiency and kinetic analysis of photocatalyst were done using the following Formulas (1) and (2):(1)$$E_{t}=1-C_{t}/C_{0}$$
(2)$$\ln\left(\frac{C_{t}}{C_{0}}\right)=-kt$$
where *E_t_* represents the degradation efficiency, *C_t_* is the concentration of the pollutant, *C_0_* represents the initial concentration of the pollutant, and *k* represents the rate constant of the photocatalytic reaction.

### 2.4. Electrochemical Measurements

The slurry of the working electrode was mixed with obtained catalyst (5 mg), isopropanol (IPA, 1 mL), and Nafion (50 μL). The working electrode was prepared by applying it on indium-tin oxide conductive film glass (ITO) and then drying it under the vacuum. The electrolyte solution, counter electrode, and reference electrode were sodium sulfate (0.5 mol L^−1^, 100 mL), platinum, and Ag/AgCl, respectively.

## 3. Results and Discussion

### 3.1. Structure and Morphology

The X-ray diffraction (XRD) patterns of BiOBr, *g*-C_3_N_4_, and x-*g*-C_3_N_4_/BiOBr (x = 4, 6, 8) are indexed to BiOBr (PDF # 09-0393) and *g*-C_3_N_4_ (Figure 1a and Appendix A). The diffraction peaks at 27.6° from the (002) plane of *g*-C_3_N_4_ were detected in the XRD pattern of *g*-C_3_N_4_/BiOBr, suggesting successful preparation of the *g*-C_3_N_4_/BiOBr [35]. The results of the Fourier transform infrared spectroscopy of the *g*-C_3_N_4_/BiOBr are shown in Figure 1b. The peaks at 515 cm^−1^, 807 cm^−1^, and 1032–1772 cm^−1^ are assigned to the stretching vibration of the Bi-O, 3-s-triazine ring structure and the C-N bond, respectively, indicating that *g*-C_3_N_4_ couples with BiOBr to form a heterojunction [42,43,44]. In addition, the wider absorption peak at 3000–3500 cm^−1^ corresponds to the stretching vibration of -OH in the water absorbed by the sample surface [14].

The morphologies of the BiOBr, *g*-C_3_N_4_, and *g*-C_3_N_4_/BiOBr are characterized using scanning electron microscopy (SEM) and transmission electron microscopy (TEM). The BiOBr and *g*-C_3_N_4_ possess petal-like and block structures, respectively (Figure 1c,d). It can be observed that the BiOBr embedded into the surface of the block of *g*-C_3_N_4_, showing a tightly connected structure (Figure 1e,f). Figure 2a–e are TEM images of BiOBr, *g*-C_3_N_4_, and *g*-C_3_N_4_/BiOBr. Notably, the distinct boundary is observed in *g*-C_3_N_4_/BiOBr (Figure 2d), implying tight contact. This tight contact structure enables the charge separation and enhances the photocatalytic degradation performance. The lattice stripe (0.28 nm) corresponding to the (110) crystal plane for BiOBr is observed (Appendix A). Additionally, the amorphous structure of *g*-C_3_N_4_ is evident as the lattice stripes are not detected in *g*-C_3_N_4_ (Appendix A), which is consistent with the result of the XRD. As seen in the inset of Figure 2f, the selective area electron diffraction (SAED) confirms that the BiOBr platelets are monocrystalline. The base plane is classified as the (001) plane due to the direction of the electron beam being perpendicular to its base plane [45]. Additionally, the presence of Bi, Br, O, C, and N is demonstrated by the TEM mapping (Figure 2g–l).

The X-ray photoelectron spectroscopy (XPS, Figure 3a) survey spectra reveal the existence of Bi, Br, O, C, and N elements in the *g*-C_3_N_4_/BiOBr. The positions of the Bi 4f (Figure 3b), Br 3d (Figure 3c), and O 1s (Figure 3d) peaks in *g*-C_3_N_4_/BiOBr shift slightly to higher binding energy compared those in BiOBr. Conversely, the binding energy of N 1s (Figure 3f) moves towards a lower binding energy position than that of the *g*-C_3_N_4_. This shift in binding energy indicates a strong interaction at the *g*-C_3_N_4_/BiOBr interface instead of a simple physical mixing, which is consistent with the TEM images.

### 3.2. Band Structure of Catalysts

The UV-visible diffuse reflectance spectra are shown in Figure 4a. The BiOBr and *g*-C_3_N_4_ display absorption edges of 445 nm and 468 nm, respectively. Obviously, the light absorption edge of the *g*-C_3_N_4_/BiOBr exhibits a red shifting to 450 nm, indicating that the optical absorption range is improved by the introduction of *g*-C_3_N_4_.

The energy band structures of *g*-C_3_N_4_/BiOBr are determined by a Tauc plot (Figure 4b) and the Formulas (S1) and (S2) [46,47]. It is calculated that the CB position of the BiOBr and *g*-C_3_N_4_ is 0.33 V and −0.94 V, and the VB position is 3.01 V and 1.59 V, respectively. According to the valence band XPS spectra (Figure 4c,d), the distance from the VB to the *E_f_* is 2.03 eV (BiOBr) and 1.65 eV (*g*-C_3_N_4_). Therefore, the *E_f_* of BiOBr and *g*-C_3_N_4_ is 0.98 V and −0.06 V, respectively. Thus, the CB (−0.94 V) and *E_f_* (−0.06 V) of *g*-C_3_N_4_ are more negative than BiOBr (CB: 0.33 V, *E_f_*: 0.98 V). As depicted in Figure 4e, when *g*-C_3_N_4_ contacts BiOBr, the difference in *E_f_* drives the electrons migrating from BiOBr to *g*-C_3_N_4_ until their Fermi levels equalize. This electronic transfer promotes the separation of carriers and enables the high redox capability of photogenerated e^−^-h^+^.

### 3.3. Photocatalytic Degradation

Photocatalytic degradation experiments are carried out under visible light irradiation. The adsorption efficiencies of BiOBr, *g*-C_3_N_4_, and *g*-C_3_N_4_/BiOBr are as slight as 5.2%, 6.6%, and 8.9%, respectively (Figure 5a). After 90 min of visible light irradiation, the photocatalytic efficiency in degrading MB is 79.6% (BiOBr), 70.5% (*g*-C_3_N_4_), and 90.8% (*g*-C_3_N_4_/BiOBr), respectively. The first-order kinetic analysis demonstrates that the photocatalytic kinetic constant of *g*-C_3_N_4_/BiOBr (26.89 min^−1^ × 10^−3^) is 1.56 and 1.98 times faster than that of BiOBr (17.22 min^−1^ × 10^−3^) and *g*-C_3_N_4_ (13.59 min^−1^ × 10^−3^), respectively (Figure 5b). This result indicates the successful construction of S-scheme heterojunctions and the important role of the tightly connected IEF at the interface in photocatalytic degradation.

It is well-known that the degradation of pollutants in a real water environment is not only challenging but also has practical value. The degradation efficiency of *g*-C_3_N_4_/BiOBr toward CR (20 mg L^−1^), TC (50 mg L^−1^), and MB (20 mg L^−1^) is 99.2%, 89%, and 90.8%, respectively, meaning an excellent practical application potential of *g*-C_3_N_4_/BiOBr (Figure 5c). Compared with other materials, *g*-C_3_N_4_/BiOBr maintains relatively good photocatalytic degradation performance (Appendix A). In addition, we degraded TC at different concentrations (2 mg L^−1^,20 mg L^−1^, and 50 mg L^−1^, Appendix A) and found that *g*-C_3_N_4_/BiOBr still maintained excellent degradation efficiency for higher and lower concentrations of pollutants.

The degradation performance is commonly affected by catalyst dosage, pH value, and water quality during the process of photocatalytic degradation. As shown in Appendix A, the photocatalytic degradation efficiency of MB increases from 80.3% (0.25 g L^−1^) to 92.2% (1.0 g L^−1^) with a higher catalyst dosage. The effect of the pH value (pH = 4–10) on degradation is tested (Figure 5d). It can be observed that the degradation efficiency remains at 80% over a wide pH range (pH = 4–10).

Photocatalytic degradation experiments in natural water quality are conducted to further investigate the potential application of the *g*-C_3_N_4_/BiOBr. The relevant parameters of natural water bodies have been listed in Appendix A. Before degrading the MB solution, the impurities are filtered out to avoid interfering with subsequent measurements. The photocatalytic degradation efficiencies of MB by *g*-C_3_N_4_/BiOBr still remained 95.9% (tap water), 97.8% (lake water), and 99.5% (river water). Thus, the change in water quality has a minimal impact on the degradation of MB (Figure 5e). These findings indicate the excellent practical application potential of *g*-C_3_N_4_/BiOBr.

To evaluate the reusability of the *g*-C_3_N_4_/BiOBr, five consecutive photocatalytic experiments are conducted by adding the same amount of recovered *g*-C_3_N_4_/BiOBr into fresh MB solution after each reaction cycle. Figure 5f exposes that the photocatalytic activity of the *g*-C_3_N_4_/BiOBr remains 80% even after five photo-degradation cycles, certifying the good cycling stability. Simultaneously, XRD patterns (Appendix A) and SEM images (Appendix A) of the catalyst after cycling show no significant change, indicating the good cycling stability of *g*-C_3_N_4_/BiOBr.

### 3.4. Photoelectric Properties

Photoluminescence (PL) and electrochemical impedance spectroscopy (EIS) are utilized to analyze the recombination of photogenerated carriers. The *g*-C_3_N_4_/BiOBr presents the lowest PL intensity (Figure 6a) compared to that of *g*-C_3_N_4_ and BiOBr, implying the efficient separation of electron and hole. In the EIS analysis (Figure 6b), the semicircle radius of *g*-C_3_N_4_/BiOBr is smaller than that of BiOBr and *g*-C_3_N_4_, indicating a lower charge transfer resistance of *g*-C_3_N_4_/BiOBr. This result demonstrates that the tightly connected S-scheme heterojunction structure is beneficial to the migration of photogenerated carriers. Subsequently, the photocurrent intensity of *g*-C_3_N_4_/BiOBr is higher than that of BiOBr and *g*-C_3_N_4_, demonstrating that more photogenerated electrons and holes are generated (Figure 6c). These results exhibit that the S-scheme heterojunction of *g*-C_3_N_4_/BiOBr significantly promotes the separation of photogenerated carriers.

### 3.5. Photocatalytic Degradation Mechanism

To verify the types of active free radicals, quenching experiments are tested (Figure 6d). Isopropanol (IPA), p-benzoquinone (BQ), and formic acid (FA) were chosen to quench ·OH, ·O^2−^, and h^+^, respectively [48,49,50,51,52]. The degradation efficiency is decreased by 8.3%, 24.7%, and 31.7% after introducing ·OH, ·O_2_^−^, and h^+^ scavenger, respectively. Therefore, in the *g*-C_3_N_4_/BiOBr degradation system, h^+^ and ·O_2_^−^ are the main active species, and the ·OH plays weakly degrades MB.

According to the results of the above characterizations and experiments, we believe that the high activity of photocatalysts comes from three aspects. Firstly, the exposure of (001) crystal planes of BiOBr provides more active sites for photocatalytic reactions. Secondly, the tightly connected S-scheme heterojunction structure provides a fast channel for photogenerated electrons (Figure 2d and Figure 6b). Finally, the construction of S-scheme heterojunctions effectively accelerates the separation of photogenerated carriers and keeps the redox ability. Therefore, the photocatalytic efficiency is significantly improved under the synergistic effect of high-activity (001) crystal planes exposure, a tightly connected structure, and S-type heterojunction.

Based on the above results, the degradation mechanism can be described as shown in Figure 7. An IEF from *g*-C_3_N_4_ to BiOBr is formed. This IEF induces the photogenerated e^−^ in the CB of BiOBr to combine with the h^+^ in the VB of *g*-C_3_N_4_, significantly promoting carrier separation under the irradiation of visible light. Thus, the retained h^+^ in the VB of BiOBr (3.01 V) can react with H_2_O/OH^−^ to form ·OH or directly react with pollutants, while the retained e^−^ in the CB of *g*-C_3_N_4_ can react with O_2_ dissolved in the solution to form ·O^2−^. Due to the tightly connected S-scheme heterojunction, effective charge transfer facilitates the generation of ·OH, ·O^2−^, and h^+^ to achieve the effective degradation of organic pollutants. The photocatalytic process is as follows (Formulas (3)–(6)):(3)$$g\text{-}C_{3}N_{4}/\mathrm{BiOBr}\;\overset{hv}{\to }e^{-}+h^{+}$$
OH^−^ + h^+^ → ·OH(4)
O_2_ + e^−^ → ·O_2_^−^
(5)
Org. (MB, CR, TC) + ROS (h^+^, ·OH, ·O_2_^−^) → CO_2_ + H_2_O (6)

## 4. Conclusions

In summary, we have prepared efficient and closely contacted *g*-C_3_N_4_/BiOBr (001) S-scheme heterojunction photocatalysts using a simple hydrothermal method. The synergy of highly active (001) crystal plane exposure, a tightly connected structure, and the construction of a S-scheme heterojunctions structure significantly enhances the photocatalytic performance of *g*-C_3_N_4_/BiOBr. The exposure of a highly active (001) crystal plane provides more active sites for photocatalytic reactions. The formation of close-contact structures improves the efficiency of electron transfer. And the construction of S-scheme heterojunctions accelerates the separation of charge carriers. Additionally, *g*-C_3_N_4_/BiOBr exhibits an excellent degradation performance for different pollutants (MB: 90.8%, CR: 99.2%, and TC: 89%), wide pH conditions (pH = 4–10), and different water qualities. This work provides a feasible strategy for preparing novel stepped photocatalysts for water treatment.

## Figures and Tables

**Figure 1 nanomaterials-14-01071-f001:**
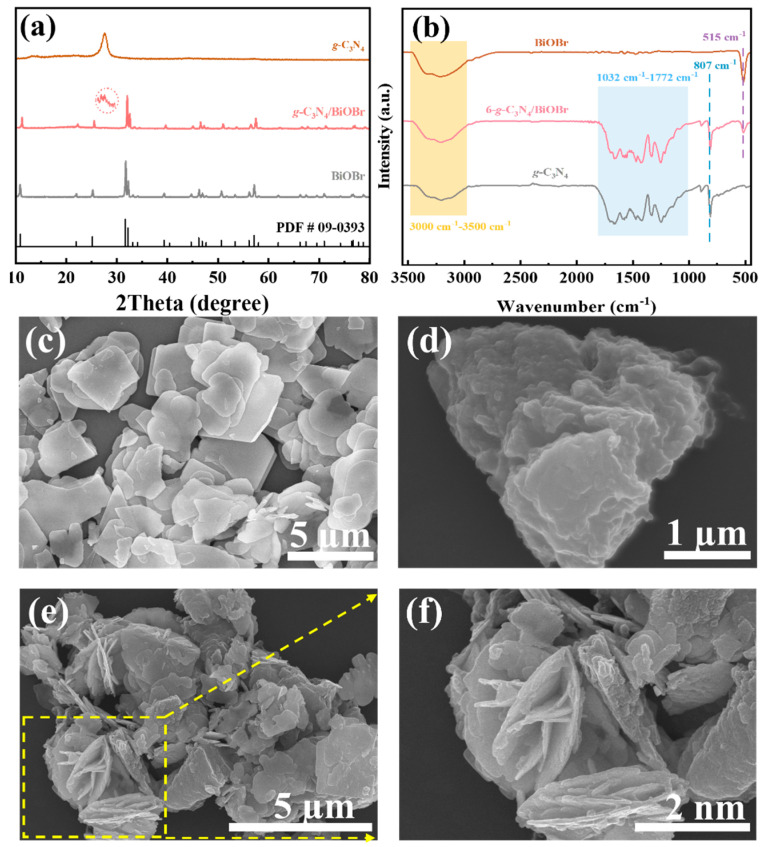
(**a**) XRD patterns; (**b**) FT-IR spectra; and (**c**–**f**) SEM images of BiOBr, *g*-C_3_N_4_, and *g*-C_3_N_4_/BiOBr.

**Figure 2 nanomaterials-14-01071-f002:**
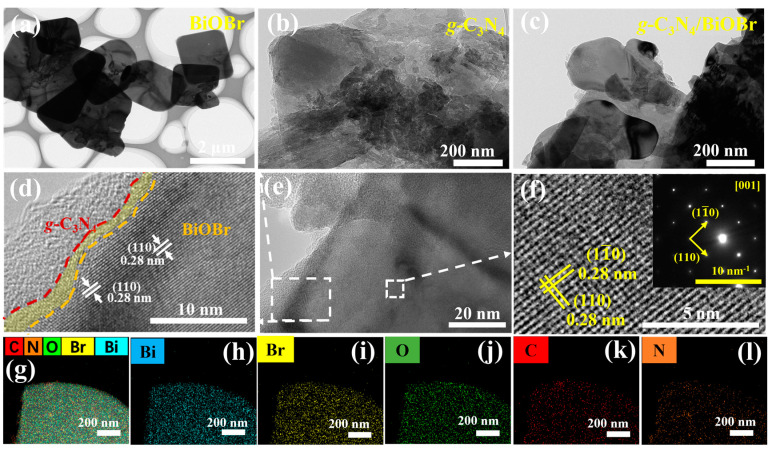
TEM images of (**a**) BiOBr, (**b**) *g*-C_3_N_4_, and (**c**) *g*-C_3_N_4_/BiOBr; (**d**–**f**) HRTEM images of *g*-C_3_N_4_/BiOBr; (**g**–**l**) TEM mapping: Bi, Br, O, C, and N.

**Figure 3 nanomaterials-14-01071-f003:**
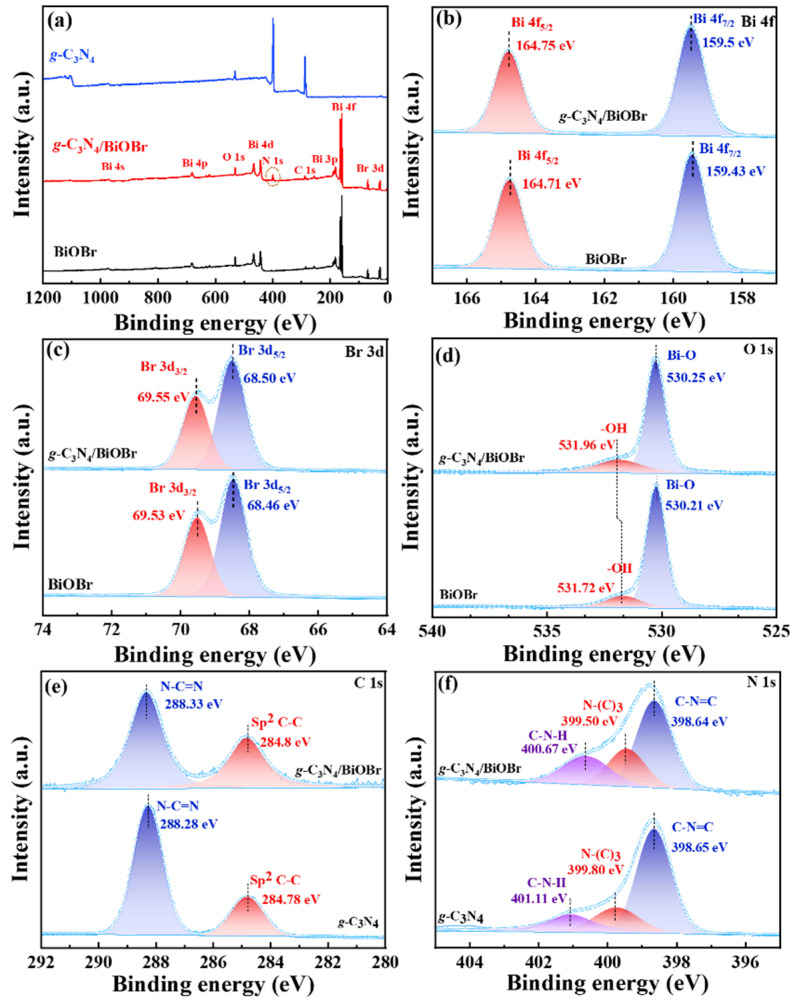
(**a**) XPS spectra for the survey, (**b**) Bi 4f, (**c**) Br 3d, (**d**) O 1s, (**e**) C 1s, and (**f**) N 1s XPS spectra of BiOBr, *g*-C_3_N_4_, and *g*-C_3_N_4_/BiOBr.

**Figure 4 nanomaterials-14-01071-f004:**
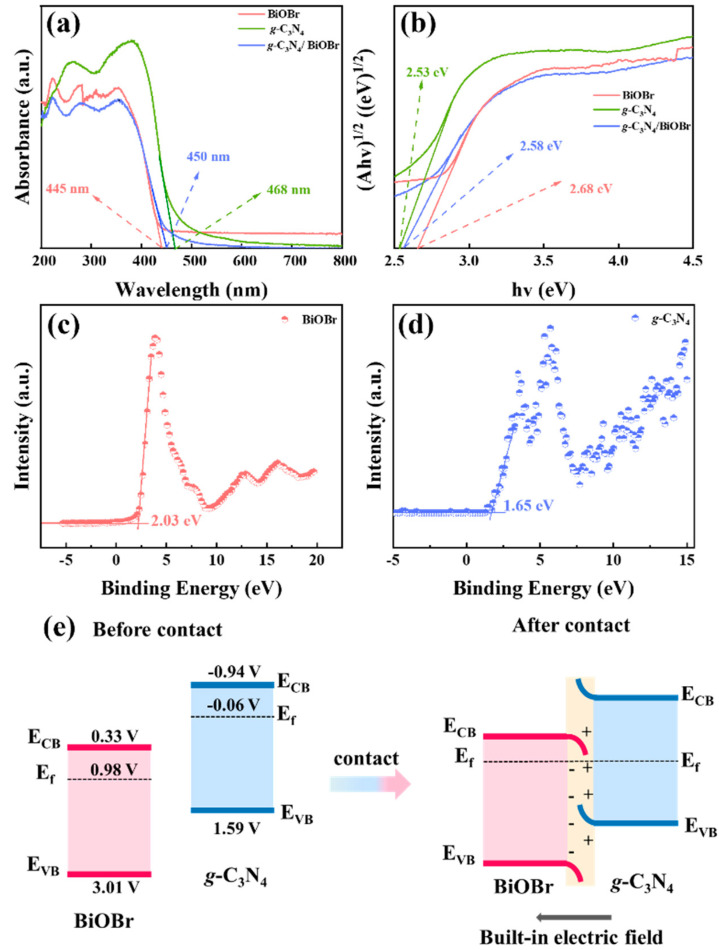
(**a**) UV-visible diffuse-reflectance spectra; (**b**) the Tauc plot of BiOBr and *g*-C_3_N_4_; (**c**,**d**) the valance band XPS spectra of BiOBr and *g*-C_3_N_4_; and (**e**) the band structure of BiOBr and *g*-C_3_N_4_.

**Figure 5 nanomaterials-14-01071-f005:**
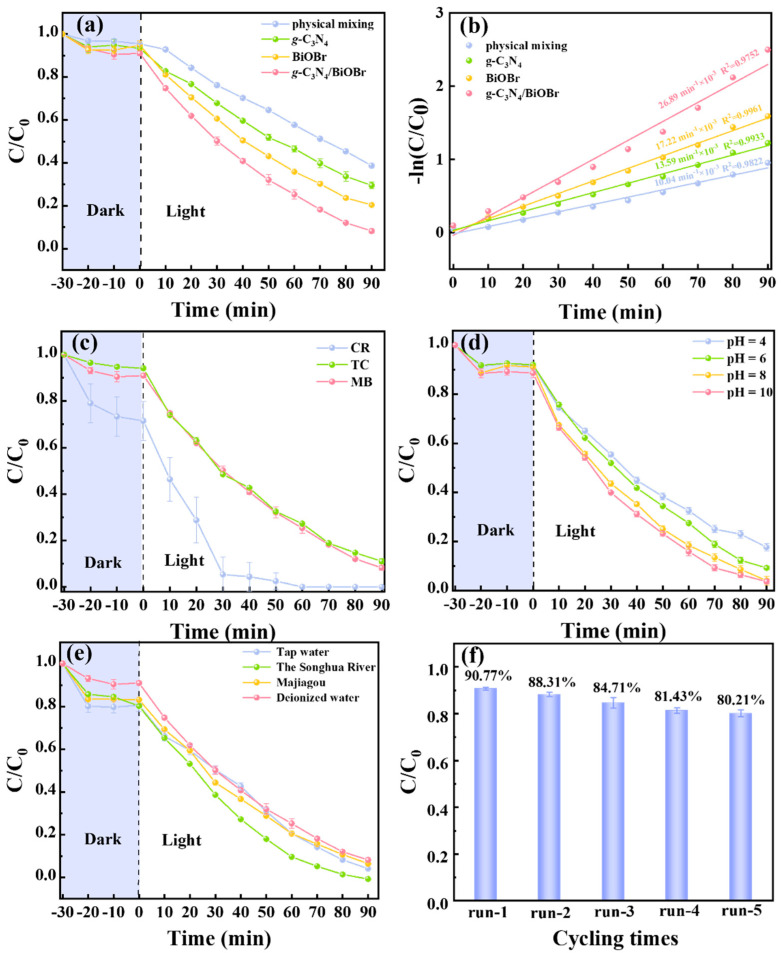
(**a**) Photocatalytic degradation efficiency of *g*-C_3_N_4_, BiOBr, physical mixed samples, and *g*-C_3_N_4_/BiOBr; (**b**) first-order kinetic constants of MB degradation; (**c**) photocatalytic degradation of CR, TC, and MB; (**d**) initial pH (4–10); (**e**) photocatalytic degradation of MB in natural water quality; (**f**) the cyclic catalytic degradation of MB in the presence of *g*-C_3_N_4_/BiOBr.

**Figure 6 nanomaterials-14-01071-f006:**
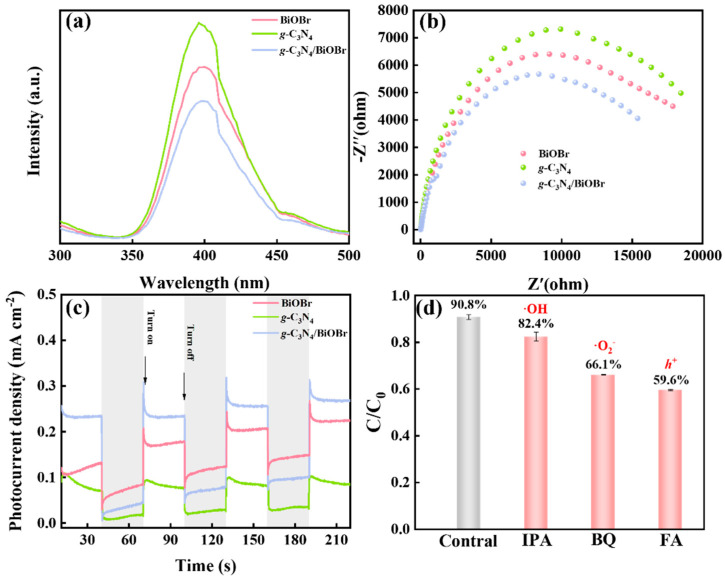
(**a**) PL spectra; (**b**) EIS spectra; (**c**) transient photocurrent responses; (**d**) quenching experiments.

**Figure 7 nanomaterials-14-01071-f007:**
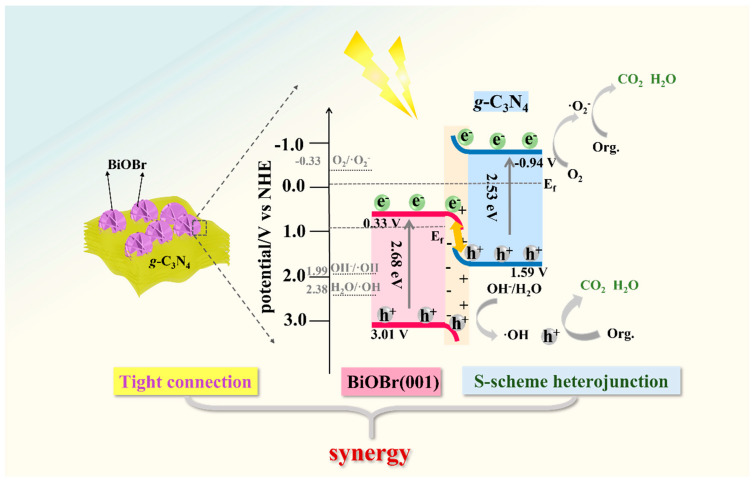
Photocatalytic mechanism of MB by the *g*-C_3_N_4_/BiOBr.

## Data Availability

Data are contained within the article.

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
