# Peer review of "A Tight-Connection g-C3N4/BiOBr (001) S-Scheme Heterojunction Photocatalyst for Boosting Photocatalytic Degradation of Organic Pollutants"

_nanomaterials, 2024, doi:10.3390/nano14131071_

Round 1

Reviewer 1 Report

Comments and Suggestions for Authors

The authors describe the synthesis and application of g-C3N4/BiOBr S-scheme materials for organic pollutants degradation in water.

Although the novelty is poor, the quality of the results merit to be published after major revisions.

1- The introduction must be improved. First of all, the limits described for BiOBr are not so drastic. Many papers report the high activity of this photocatalyst. The bandgap is "bandgap" and not "gap band". The percetages are not weight ratios. The state of the art must be reported and the authors must explain how they overcome the limits of other materials described in the literature with those here reported.

2- In my opinion, the photocatalytic tests' description should be reported in the main text and not in the SI.

3- Results. The real composition of the synthesized materials must be reported and not only the theoretical one.

The xrd peak of g-C3N4 in the composites seems part of the background. The FTIR spectra must be recollected because they seem saturated.

Why for TC degradation was used a higher concentration of pollutant ?

The authors report the results obtained using different types of water matrices. Were the tap and lake water matrices synthetic (prepared in lab) or real matrices?

In any case, the composition must be investigated.

In fact, the kinetic behaviour of the photocatalyst in lake water matrix is not linear and the reasons must be reported.

Comments on the Quality of English Language

A revision is required

Reviewer 2 Report

Comments and Suggestions for Authors

- English language must be improved;

- Keywords are missing;

- Objectives and novelty of the work must be clarified;

- The selection of the targeted pollutants must be justified as well as the concentrations applied;

- Section 2.1.3 is not very clear and must be rewritten with further details. I believe that g-C3N4 was not dissolved but suspended in water. The preparation of the heterojunction must be made clear.

- The description of several analytical and reaction procedures are missing such as the way the catalysts were characterized; how was the photocatalytic oxidation carried out; how was the water quality assessed.

- The characteristics of the natural water used as matrix must be given.

- Figure 5 is not totally clear: what do you mean about physical mixing? What do you mean about photocatalytic degradation of different catalysts?  First-order kinetic constants for the degradation of what? The caption must be clear and self sustained with all the conditions. Moreover, this figure deserves further discussion.

Comments on the Quality of English Language

English language requires improvement for a better understanding.

Round 2

Reviewer 1 Report

Comments and Suggestions for Authors

The authors have improved their manuscript. However, before the publication a few errors/comments must be corrected.

Table S1. ICP of 4-g-C3N4/BiOBr, 6-g-C3N4/BiOBr and 8-g-C3N4/BiOBr: in column 2 Bi3+ and not Bi+.

The explanation introduced to justify the lower concentration of TC in the degradation tests is not convincing. Please, improve this aspect.
